# Evaluating Weakly Supervised Object Localization Methods Right? A Study on Heatmap-based XAI and Neural Backed Decision Tree

## Abstract

Choe et al have investigated several aspects of Weakly Supervised Object Localization (WSOL) with only image label. They addressed the ill-posed nature of the problem and showed that WSOL has not significantly improved beyond the baseline method class activation mapping (CAM). We report the results of similar experiments on ResNet50 with some crucial differences: (1) we perform WSOL using heatmap-based eXplanaible AI (XAI) methods (2) we consider the XAI aspect of WSOL in which localization is used as the explanation of a class prediction task. Under similar protocol, we find that XAI methods perform WSOL with very sub-standard MaxBoxAcc scores. The experiment is then repeated for the same model trained with Neural Backed Decision Tree (NBDT) and we found that vanilla CAM yields significantly better WSOL performance after NBDT training.

## 1 Introduction

Weakly-supervised object localization (WSOL) aims to use only image-level labels (class labels) to perform localization. Compared to methods that require full annotations, WSOL can be much more resource efficient; it has therefore been widely studied (Choe & Shim, 2019; Singh & Lee, 2017; Zhang et al., 2018a;b; Zhou et al., 2016; Guo et al., 2021; Wei et al., 2021; Babar & Das, 2021; Gao et al., 2021; Xie et al., 2021).

Class Activation Mapping (CAM) (Zhou et al., 2016) is a heatmap-based explainable artificial intelligence (XAI) method that enables Convolutional Neural Network (CNN) to perform WSOL. Other heatmap-based XAI methods have been designed to compute relevance/attribution maps, some of which have been treated as localization maps after some processing e.g. Saliency (Simonyan et al., 2014) has been used for WSOL using only gradient (obtained from backpropagation) and minimal post-processing. In this paper, besides Saliency, we will also investigate the WSOL capability of several heatmap-based XAI methods: GradCAM (Selvaraju et al., 2016) (generalization of CAM), Guided Backpropagation (GBP) (Springenberg et al., 2015) and DeepLift (Shrikumar et al., 2017). Admittedly, there are many other methods that are not included in this paper e.g., Layerwise Relevance Propagation (also its derivatives (Bach et al., 2015; Montavon et al., 2017; Kohlbrenner et al., 2020)) and modifications of CAM (Muhammad & Yeasin, 2020; Wang et al., 2020; Jalwana et al., 2021; Kindermans et al., 2018).

**Main objective of this paper**: measure the WSOL capability of existing heatmap-based XAI method applied on ResNet50 and improve them. Fig. 1 shows how existing XAI methods can be very unsuitable for WSOL (e.g. high granular heatmaps and uneven edge detection). This paper shows that it is possible to modify the aforementioned methods and improve their localization ability beyond baseline CAM. *Important clarifications*:

1. It is not our intention to invent yet another XAI method. Instead, we add intermediate steps (based on CAM-like concept) on existing techniques to improve WSOL performance.
2. We do not claim to attain the best localization. We are in fact testing the metric MaxBoxAcc presented in CVPR 2020 (Choe et al., 2020). In that paper, a dedicated training process is performed to optimize the said metric; in their github, this training is simply called the *WSOL training*. While their training improved WSOL, as a trade-off, their classification performance has degraded, and we quote them "There are in general great fluctuations in the classification results

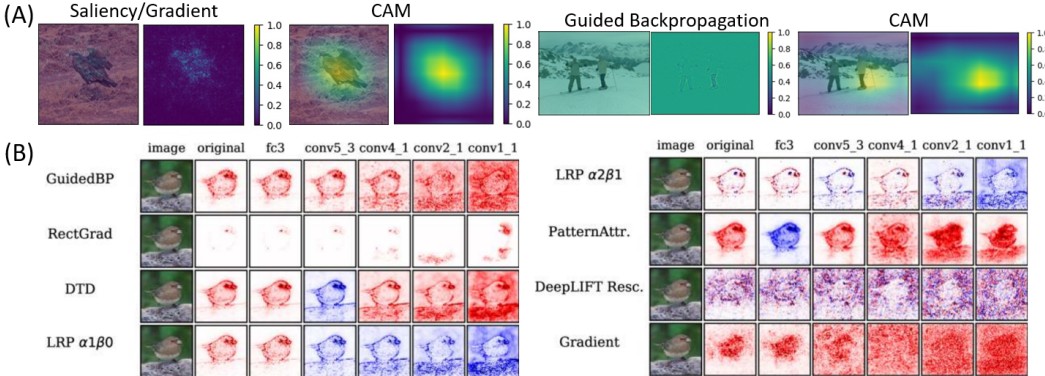

Figure 1: Localization capabilities of XAI methods differ, some may not be suitable for WSOL (A) When naively normalized to $[0, 1]$, saliency method shows sparse and granular heatmaps compared to CAM, while guided BP shows an attribution map with unclear localization. (B) Heatmaps from various methods during sanity checks. We can infer that different localization information may (but not necessarily) exist within different layers. Figure (B) is used with permission obtained from the authors of (Sixt et al., 2020).

(26.8% to 74.5% on CUB)". This accuracy degradation can be found in their appendix, section C2. By contrast, we prioritize interpretability, hence our baseline is CAM *without* WSOL training. Instead of WSOL training, we use NBDT training (see later). Other XAI methods are tested on the same metric and compared to CAM.

Summary of our contributions and results:

1. Vanilla CAM after Neural Backed Decision Tree (NBDT) training yields the highest performance, besting most other methods by a significant margin. Heatmaps derived from Saliency method applied to the input layer obtains high scores as well but the method requires a peculiarly low operating threshold.

2. With the proper application of CAM-like concepts, heatmaps obtained from the inner layers of existing XAI methods can perform WSOL that beats the original CAM without NBDT.

3. The NBDT original paper (Wan et al., 2021) trains larger models from scratch. However, we successfully fine-tuned pre-trained ResNet50 without causing the collapse of predictive performance.

## 2   RELATED WORKS AND LITERATURE REVIEW

Our main references are Choe's paper (Choe et al., 2020) (CVPR 2020), the NBDT paper (Wan et al., 2021) (ICLR 2021) and the following heatmap-based XAI methods: CAM (as the baseline), GradCAM, Guided BP, Saliency and DeepLift.

**CAM: weighted sum of feature maps**. To use a CNN for prediction, usually the last convolutional feature maps $f \in \mathbb{R}^{(C,H,W)}$ is average-pooled and fed into fully-connected (FC) layer, i.e. $FC(AvgPool(f))$ where $C = 3$ is the color channel, $H$ height and $W$ width of the images. Suppose our CNN classifies images into $K$ different categories. Denote the $i$-th feature map before pooling by $f_i$ where $i = 1, \ldots, c$, $c$ is the number of output channels before FC. The FC weight is then given by $w \in \mathbb{R}^{c \times K}$. To obtain CAM for class $k \in \{1, \ldots, K\}$, compute weighted-sum across channel $c$ of $f$, so that $CAM = \Sigma_i^c f_i w_{ik}$. In this paper, we tested various weighting schemes i.e. different $w_{ik}$ and other empirical modifications aimed to yield better WSOL performance.

**WSOL Metric**. Popular metrics to evaluate WSOL are Top-1, Top-5 localization (accuracy or error) and GT-known localization accuracy (Choe & Shim, 2019; Singh & Lee, 2017; Zhang et al., 2018a;b; Zhou et al., 2016; Guo et al., 2021; Wei et al., 2021; Babar & Das, 2021; Gao et al., 2021; Xie et al., 2021). The problems with these simple metrics are well described in (Choe et al., 2020). Firstly, WSOL with only image-level labels can be an ill-posed problem. Secondly, the dependence on operating threshold $\tau$ may lead to misleading comparison. Thus they introduced

$MaxBoxAcc(\delta) := max_\tau BoxAcc(\tau, \delta)$ where:

$$BoxAcc(\tau, \delta) = \frac{1}{N} \sum_n 1_{IoU\left(box(s(\boldsymbol{X}^{(n)}, \tau)), B^{(n)}\right) \geq \delta} \tag{1}$$

where $\tau, \delta$ are the score map (operating) threshold and IoU threshold respectively. $B$ is the ground-truth box and $box(s(\boldsymbol{X}^{(n)}), \tau)$ is the tightest box surrounding the connected component with max area of mask $\{(i,j)|s(X_{ij}^{(n)}) \geq \tau\}$. Note: Opencv has been used to obtain $box(.)$ (contour algorithm, boundingRect etc).

Notice that $\delta$ is chosen as the threshold that yields the most number of "passing" IOU ($> \delta$) *across all N data*: in the original paper's python code, the values are "accumulated" over all samples before computing the final score. The consequence is clear: the best $\tau$ is chosen in a discrete hit and miss manner so that a heatmap that is perfectly good (high IoU) at a non-chosen threshold *might be considered a miss* in the chosen threshold. Remark: after publication, (Choe et al., 2020) introduced the improved version MaxBoxAccV2 such that (1) multiple thresholds $\delta = \{0.3, 0.5, 0.7\}$ are used and their results averaged (2) max area assumption is removed so that the "best match" with ground-truths are found.

In (Choe et al., 2020), models are fine-tuned specifically to improve WSOL performance through the *WSOL training*. However, fine-tuning is done at the expense of predictive performance i.e. their classification accuracy drops. **We avoid this**. Since we are interested in the XAI aspect of a model, we strive to achieve localization-based explanation for all class label predictions. As such, we attempt to simultaneously maintain the model's predictive power as we perform WSOL. The NBDT architecture appears to be a promising choice to achieve this.

**NBDT** is originally devised to improve not only the predictive power of a CNN, but also its interpretability through the inherent structure of a decision tree. Although it does not strictly improve the predictive power (sometimes the accuracy drops slightly, as shown in NBDT paper and here), we bring NBDT one step further by demonstrating its additional benefit: NBDT can improve the WSOL capability of a model through the application of heatmap-based XAI, particularly (perhaps surprisingly) through the vanilla CAM.

NBDT inference proceeds in three steps. **Step 1**. DNN FC weights $w \in \mathbb{R}^{D \times K}$ is used to seed decision tree nodes, where $K$ is the no. of classes: for leaf nodes $i$, $n_i = w_k$ where $i = k \in \{1, 2, \ldots, K\}$ i.e. node weight is seeded with row vectors of $W$; for all inner nodes $i \in \{K+1, N\}$, $n_i = \sum_{k \in L(i)} w_k / |L(i)|$ where $k \in L(i)$ are leaves in the subtree of $i$. **Step 2**. For each sample $x$, compute probability of the child $j \in C(i)$ of node $i$ as $p(j|i) = softmax(\langle \vec{n}_i, x \rangle)[j]$ where $\langle ., . \rangle$ is the usual dot product. **Step 3**. We only use soft inference (which is shown to be the best in NBDT paper), so the final class prediction is $\hat{k} = argmax_k p(k)$ where $p(k) = \Pi_{i \in P_k} p(C_k(i)|i)$ and $p(C_k(i)|i)$ is the probability of each node $i$ in path $P_k$ traversing the next node in path $P_k \cap C(i)$.

To obtain the tree, *induced hierarchy* is needed. In NBDT, the hierarchy is constructed using FC weights $w$ of a pre-trained model plus agglomerative clustering algorithm on $w_k/||w_k||_2$; as before $w_k$ is also the row vector of $w$. The weights are paired, and then constructed into binary trees whose leaves correspond to the classes, i.e. $k = \{1, \ldots, K\}$.

Finally, fine-tuning is performed by optimizing a standard cross entropy loss plus the *soft tree-supervision loss* over class distribution of path probabilities $\mathcal{D}_{nbdt} = \{p(k)\}_{k=1}^K$, so

$$\mathcal{L} = \beta_t CELoss(\mathcal{D}_{pred}, \mathcal{D}_{label}) + \omega_t CELoss(\mathcal{D}_{nbdt}, \mathcal{D}_{label}) \tag{2}$$

where CELoss denotes cross-entropy loss, $\beta_t, \omega_t$ are supervision weights and decay factor respectively. Details are in the code (supp. material).

## 3 METHODS

As our starting point, we perform WSOL evaluations on GradCAM using MaxBoxAcc on 4 different layers within ResNet50. GradCAM on layer 4, or vanilla CAM, is used as the baseline. We then repeat the same process for 3 other different XAI methods, each also on 4 different layers. Trial and errors are necessary during our experiments because some heatmaps are not visually sensible. In the next stage, we repeat the entire process on ResNet50 after NBDT training. The entire pipeline is

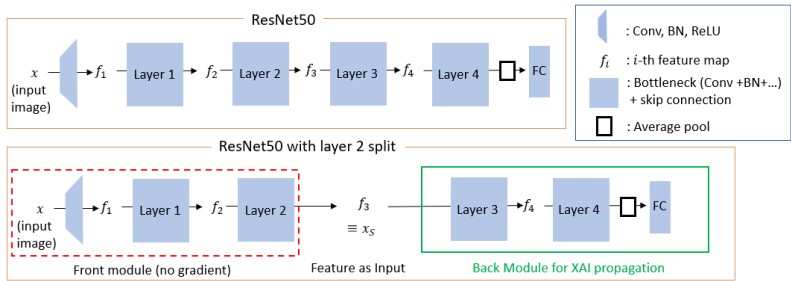

Figure 2: Layer splitting on ResNet50. Unlike GradCAM, some XAI methods are applicable only to the input layers in pytorch Captum implementation. We perform layer splitting to allow the application of other XAI methods on different layers.

resource intensive as we experiment on multiple combinations of methods and layers, thus, in this paper, we report only the result for ResNet50 and leave other models for future work. The ImageNet dataset is used for training and evaluation (same setup as the original NBDT code).

**CAM-inspired sum of weighted feature maps**. In the literature review, we saw that CAM can be easily applied to the latest convolutional layer since there are $c$ feature maps and also $c$ rows in $w$, the weight of the final FC layer. However, different localization information might also be present in different layers within a deep CNN (see fig. 1(B)), hence we explore the effect of weighted sum of feature maps on inner layers.

GradCAM has been introduced as a generalization of CAM which extends the computation of CAM to layers other than the final convolutional layer. With the LayerGradCAM module of pytorch Captum, *layerwise heatmaps* can be extracted directly. For other methods, to obtain similar heatmaps, we perform *layerwise splitting* (see below). Once layerwise heatmaps are generated, *heatmap reformat* (see below) is performed. This includes the normalization of heatmaps and the application of aforementioned CAM-like sum of weighted feature maps with empirical choices of weights. Additional modifications are applied based on empirical observations and trial and error. For example, when we observe granular heatmaps, we apply average pooling to "patch up" the empty spaces.

*Layer splitting*. Fig. 2 illustrates splitting done at layer $l = 2$. Splitting at other layer $l$ is done similarly as the following. Step 1. Propagate input $x$ through the *front module* without any gradient, where front module is the successive layers in the deep NN up to and including layer $l$. The output is denoted as $x_S = NN_{front}(x)$; here NN is ResNet50. Step 2. Perform heatmap/attribution computation as though *back module* is the entire neural network with $x_S$ as the input, i.e. heatmap $h = attr(NN_{back}, pred)$ where $pred$ is the class prediction and $attr =$ DeepLIFT, GBP or Saliency.

*Heatmap reformat* is the post-processing phase with empirical settings. All heatmaps are resized to input size $(H, W)$ and normalized to $[0, 1]$. By *recentre* we mean $x - min(x)$.

1. For GradCAM, the resulting heatmaps have only 1 channel, hence they only need to be resized, recentred and then normalized.

2. For Saliency applied to the input layer, we follow the original paper: for each pixel, take max over all heatmap channels and then take absolute values. We then normalize and resize it accordingly. For Saliency applied to inner layers, modified CAM concept is applied with pooling i.e. heatmaps are obtained through the application of 2D average pooling on weighted feature maps $avgpool(\Sigma_i^c w_{ik} f_i)$ where weights are given by the total channel magnitude $w_{ik} = \Sigma_{h,w} f_{i,h,w}$ regardless of class $k$. We found that these steps yield positive effects on WSOL performance, allowing them to beat the baseline.

3. For DeepLIFT, sum across channels (i.e. weighted sum with $w_{ik} = 1$ for all channels) yield positive performance. Several choices of variable weights on DeepLIFT did not yield satisfactory performance during our trial and error.

4. GBP on input layer is similar to DeepLIFT, but GBP on layer 1,2 and 3 yield interesting inverted heatmaps before processing. Hence, we perform the following: (1) invert heatmap $h \leftarrow 1 - h$, (2)

Table 1: **WSOL Evaluation relative to CAM without NBDT**. Vanilla CAM after NBDT yields the best score but other methods beat the CAM baseline too. More significant improvements/degradation in MaxBoxAcc are highlighted green/red. Heatmaps/attributions are generated w.r.t layer $l$, see *layer splitting*. Note: $l = 0$ means input. These scores are averaged across different $\delta$ values (including the low scoring $\delta = 70$).

| ResNet50 | CAM (baseline) MaxBoxAccV2.1 = 2.143 | | | | | | |
|---|---|---|---|---|---|---|---|
| | No NBDT (MaxBoxAccV2.1) | | | | NBDT (MaxBoxAccV2.1) | | |
| $l$ | GradCAM | Saliency | DeepLIFT | GBP | GradCAM | Saliency | DeepLIFT | GBP |
| 0 | +0.390 | +1.388 | -0.828 | -0.856 | -0.040 | +2.073 | -0.841 | -0.860 |
| 1 | +0.602 | -0.001 | -0.352 | +0.490 | +0.446 | +0.144 | -0.421 | +0.355 |
| 2 | +0.921 | +0.545 | +0.742 | +1.054 | +0.716 | +0.636 | +0.640 | +1.097 |
| 3 | +0.911 | +0.152 | +0.764 | +0.252 | +0.625 | +0.134 | + 0.571 | +0.294 |
| 4 | 0 (is CAM) | N.A. | +0.002 | N.A. | **+2.417** | N.A. | +0.216 | N.A. |

apply sum across channels, (3) apply 2D maxpool before we finally recentre, normalize and resize the heatmaps. With these, GBP in these layers performs better than the baseline.

We also report N.A. results in table 1. This means the heatmaps obtained are not making sense, for example, heatmaps with all zeros.

**WSOL Metric**. Once we have determined how to modify the XAI methods for a more visually sensible localization, we will quantitatively measure their WSOL performances. We use MaxBoxAccV2.1, which is simply MaxBoxAccV2 of (Choe et al., 2020) but with the original assumption (2) reapplied. The assumption was: the *largest* connected component is chosen for box estimation. Our rationale for using the assumption: a good XAI method is expected to yield accurate heatmaps/attributions in which large noisy patches should be suppressed or not exist altogether i.e. there is no large artifact. The most salient box for an accurate heatmap is thus already the largest component area. Furthermore, when deployed for application, ground-truth box is not necessarily available, hence there may not exist "best match" for MaxBoxAccV2 to use in practice; by contrast, there is always a largest component as long as the heatmap is not too uniform.

**NBDT training and then WSOL evaluation**. As previously mentioned, we will repeat the entire WSOL evaluation process, but this time with NBDT training process. Our experiments did not require retraining of ResNet50 from scratch. Our settings are nearly identical to the original NBDT paper with some differences as the following. We use 256 batch size (split across 8 NVIDIA Tesla V100 GPUs) and disable learning rate scheduling, using instead a constant 0.001 learning rate. No mode collapse occurred throughout all training epochs. Each epoch run on the entire ImageNet training dataset. After NBDT training, the entire MaxBoxAccV2.1 measurements are repeated for all aforementioned XAI methods and layers. Note that heatmaps are generated without the embedded decision rules for soft inferences, i.e we reuse the FC layer (there has yet to be a way to propagate signals for XAI methods through the decision tree).

We collect the result (out of 14 epochs) from the best performing checkpoint. Results are only taken after at least 10 epochs of training (see appendix table 3) since we want to make sure that the effect of soft tree-supervision loss on WSOL is present.

## 4 EXPERIMENTS AND RESULTS

**MaxBoxAcc is a stringent metric**. Let the operating thresholds be $T_{cam} = \{\tau_k \in [0,1) : k = 1, \ldots, 100\}$; this is also called *CAM interval* in the original code of (Choe et al., 2020), implemented as numpy.arange(0,1,$d_{cam}$) where $d_{cam} = 0.01$ is the cam_curve_interval.

Based on equation (1), *MaxBoxAcc* considers a bounding-box "correct" if the IoU exceeds threshold $\delta$. This is a very stringent condition since a single "best" operating threshold is used for all images. To score high on this metric, the collection of heatmaps need to fall into a particular form of distributions, as the following. Let the subset of operating thresholds used to extract the correct tight bounding-box from a heatmap be $T_n = \{\tau_k \in T_{cam} : IoU(s(...), \tau_k) \geq \delta\}_n$ where $n = 1, \ldots, N$ indexes a

particular image. Then, to obtain a high score, qualitatively, *many images need to share the same interval thresholds*.

More precisely, there has to exist an element $\mathcal{T}$ in the power set $\mathcal{P}\{T_n : n = 1, \ldots, N\}$ such that $|\mathcal{T}|$ approaches $N$ and the intersection of elements in $\mathcal{T}$ is *strictly non-empty*. This appears to be a very stringent condition to achieve: as a gauge, throughout the paper, the best score obtained is 12.8 (which is +10.688 from baseline average), achieved by CAM after NBDT at $\delta = 30$. In other words, only 12.8% samples are localized correctly. The following are the advantages and disadvantages of using MaxBoxAcc for further considerations.

*Advantages*. MaxBoxAcc can be very useful:

1. Methods that score high on this metric are useful in practice mainly because there is a consistent window of operating threshold that will yield good bounding-boxes. We do not need to worry about selecting different "correct threshold" each time during application.
2. *Anti extreme-thresholding artifact*, see next sub-section.

*Disadvantages*. Vanilla CAM scores very low on this metric although it has been considered successful in WSOL. We obtain a score of only 2.143 for CAM baseline (see table 1) averaged across $\delta = 30, 50, 70$. This means that, on average, only 2% of the entire dataset tested is considered correct: given the best $\tau$ particularly chosen through MaxBoxAcc, most IoU does NOT exceed $\delta$. From fig. 4 last row, the scores for $\delta = 70$ show that WSOL break down severely at high threshold. Hence, for XAI, the averaging of thresholds recommended by (Choe et al., 2020) may not be suitable. For further considerations:

1. Before the *WSOL training* in (Choe et al., 2020), it is well-known that existing pre-trained CNN models score well on class-labeling accuracy (typically after finetuning). However, their performances on MaxBoxAcc have not been reported. In this work, we fill the gap in the literature and show that the scores are very low.
2. After *WSOL training*, WSOL performance improves significantly: we must mention that the MaxBoxAcc scores for their trained models are very high, reaching around 60 in (Choe et al., 2020). However, final class label prediction accuracy might have degraded as shown in section C2 of their appendix. To again quote one of their results, "there are in general great fluctuations in the classification results (26.8% to 74.5% on CUB)."

While we list them as *disadvantages*, we hope that the result might instead encourage more efforts to improve WSOL based on this metric.

**Improving WSOL beyond baseline**. Here, XAI methods are used to extract localization maps from the inner layers of ResNet50. With our empirical adjustments, they are able to score better than CAM on MaxBoxAccV2.1 metric. Table 1 left (No NBDT) shows the improved results of localization that we have obtained with only a few steps of post-processing. The original Saliency method appeared to score the highest (amongst others with no NBDT), attaining +1.388 relative to baseline and high spike (with peak near 10) at $\delta = 30$ in fig. 4. Other methods are able to achieve above baseline scores as well, especially when XAI methods are applied to $l = 2$ layer.

*Comparison with visual perception*. Higher scoring heatmaps might not appear to correspond well with human perception of localization, as seen in fig. 3(B) and (C). To understand this, readers should revisit MaxBoxAcc and equation (1) in details. Note that for an image, each bounding box is obtained using a "best" threshold $\tau$ i.e. MaxBoxAcc selects *only one* of all the cascading boxes (see fig. 3(B,C) last row) whose $\tau$ gives the best score after taking into account every image in the dataset. This $\tau$ could work well with many other images but is not necessarily good on the particular image being inspected, hence the localization might appear visually poor.

*Tighter cascading bounding boxes better?* Fig. 3(B) also seems to indicate that a set of cascading bounding boxes that are tighter around the ground-truth bounding boxes score better than the baseline; see fig. 3(A) for a simpler illustration. For example DeepLIFT's bounding boxes score better than CAM even though CAM's heatmaps appear visually better. Why? Given a "best" $\tau$, there is naturally a higher probability that a good bounding-box being eventually selected comes from one of the tighter bounding box *if most bounding boxes are concentrated around the ground-truth*. CAM heatmaps appear to localize very well, but there is a lack of precise, tight bounding. Likely this is the reason why other XAI methods score better than the baseline CAM under MaxBoxAcc metric. The above suggestions are based on the observations of multiple images; we show only a few images here.

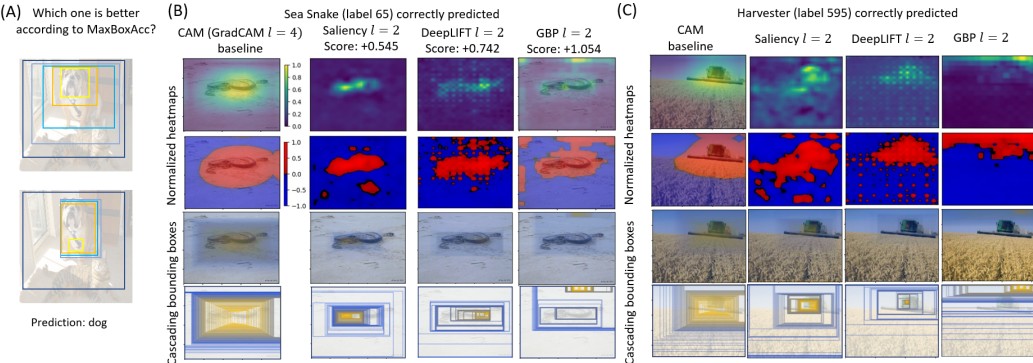

Figure 3: Heatmaps and cascading bounding boxes (CBBs). (A) Top: CAM-like CBB. Bottom: tighter CBBs appears to score better on MaxBoxAcc. (B) Row 1 and 2 show heatmaps at normal and extreme colouring thresholds. Row 3 are CBBs obtained from row 4 with each rectangle filled with colour (with high alpha transparency). Row 4 shows CBBs: blue bounding boxes are obtained from lower threshold $\tau$, yellow from higher threshold. MaxBoxAccV2.1 scores of each method is listed on the top. Recall: the score is not computed for each individual image, but an average over all images. (C) Similar to (B).

Readers must understand that the heatmap of a single image is not representative of the final score, especially due to the fact that the metric is computed over all $N$ in equation (1).

*Anti extreme-thresholding artifact.* A special benefit from the way MaxBoxAcc is defined is that we have an anti extreme-thresholding artifact, best illustrated by fig. 3(B) GBP $l = 2$. At very high thresholds, artifacts might be mistakenly selected by contour algorithm, and the results are small yellow boxes in seemingly random places. However, if the large majority of bounding boxes are tightly overlapping around the object's ground-truth bounding boxes, MaxBoxAcc's selection of $\tau$ has a higher chance of picking up a more sensible bounding box that corresponds to the ground-truth, as suggested in the previous paragraph.

**Performances at different $\delta$ thresholds**. We conducted multiple experiments across different XAI methods and layers. The results are also presented as BoxAcc vs operating threshold plots, similar to the fig. 5 of (Choe et al., 2020) for ImageNet. Our results are shown in fig. 4 (dotted lines for non-NBDT results). As before, the baseline is CAM i.e. the green dotted line of GradCAM at $l = 4$. Results at $\delta = 30$ without NBDT are shown below:

1. the original saliency attains a very high score at peculiarly low $\tau$ threshold (blue high narrow spikes). From fig. 1(A), Saliency heatmaps appear very granular, although we can see that they correspond relatively well to the shape of the object. Fig. 4 shows that within a small range of low thresholds, Saliency bounding boxes are localizing more accurately than other XAI methods.

2. GradCAM appears to perform better at $l = 2, 3$. Layer 2 curve is narrower and slightly leaned to the right compared to layer 3, i.e. there is a smaller window of slightly higher thresholds for layer 2 perform WSOL better.

3. DeepLIFT curves are less predictable. At input layer and early layers, DeepLIFT performs very poorly. However, similar to GradCAM, a relatively better localization power is found in the inner layers $l = 2, 3$, consistent the averaged results shown in table 1 (red highlight, $l = 0, 1$).

4. GBP is equally unpredictable. GBP at deeper layers show wider curves as well. The performance at lower thresholds in $l = 2$ appears to be the best.

In general, $l = 2$ seems to perform relatively better across all methods. As previously mentioned, the metric is very stringent, and nearly all the scores are below 10.

At IoU threshold $\delta = 50$, the patterns are similar. Naturally, lower scores are attained since the condition for the correctness of a bounding box is more stringent. At the highest $\delta = 70$, the results appear to have broken down into irregular hits and misses, resulting in jagged curves and very low

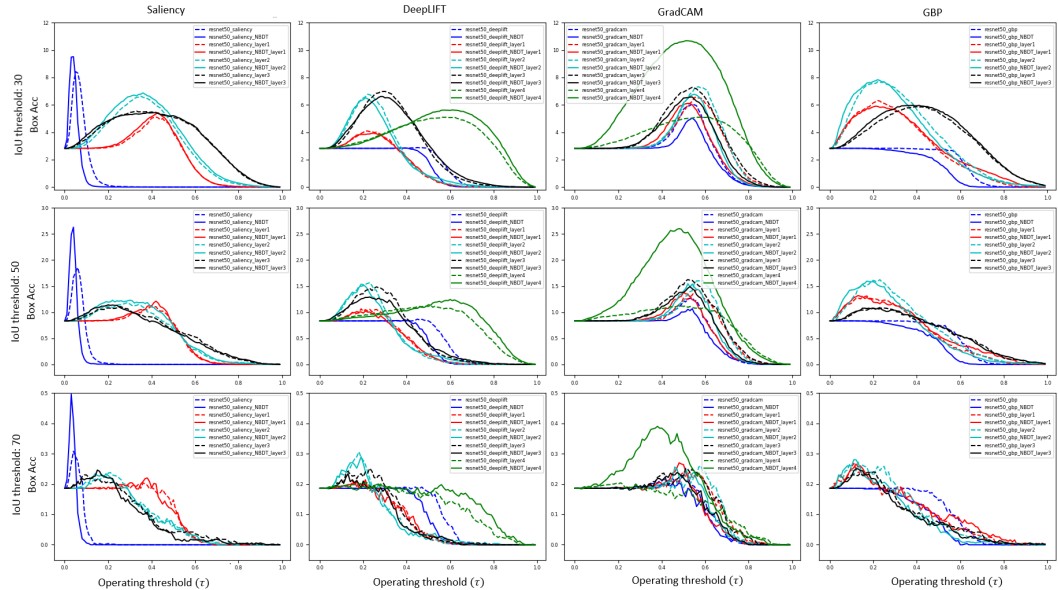

Figure 4: Performance (BoxAcc) at varying thresholds ($\tau$) plotted against the same vertical axis per IoU threshold $\delta$. XAI method, layers and post-NBDT result are indicated in the legends; when no layer is indicated, the XAI method is applied on the input. Top/Middle/Bottom are results for $\delta = 30, 50, 70$ respectively. Dotted-lines are plots for non-NBDT WSOL, solid lines for WSOL after NBDT training. The overall highest score is attained by CAM (GradCAM $l = 4$) after NBDT training. This image is best viewed electronically, zoomed in.

accuracies. The recommendation to use high $\delta$ for averaging seems unsuitable for most of these methods.

**After NBDT: training with Soft Tree-Supervision loss**. The entire experiment above is repeated after NBDT-based training is applied. Recall that our objective is to improve WSOL (better MaxBox-Acc) while maintaining predictive accuracy. NBDT has the benefit of improved interpretability from its decision tree structure.

The baseline method CAM **improved significantly** after NBDT training with an average of +2.417 improvement compared to pre-NBDT model (see table 1). The best improvement is achieved on CAM at $\delta = 30$, with a remarkable +10.688 increase. Furthermore, for all thresholds, post-NBDT CAM show the widest and tallest BoxAcc curves, as shown in fig. 4 green solid line (GradCAM $l = 4$); it is in fact the best performing method throughout our experiments.

The overall result is mixed when we include other XAI methods into consideration. Most results beat the pre-NBDT baseline CAM although some WSOL performances only marginally improve e.g. for GBP at $l = 2$, the score is $1.097 > 1.054$. The scores generally remain low, and their improvements are sometimes lower than pre-NBDT models e.g. DeepLIFT for $l = 0, 1, 2, 3$ with NBDT are all worse than pre-NBDT results. In general, other than CAM, the only other visible change after NBDT is Saliency at $l = 0$ (original version). Across all IoU thresholds $\delta = 30, 50, 70$, its MaxBoxAccV2.1 performance increases even higher compared to other XAI methods. Its best threshold $\tau$ appears to shift to an even lower value, hence a taller blue spikes in fig. 4 with peaks that are nearer to 0 along the horizontal axis.

In this section, ResNet50 has undergone 13 epochs of training and the 12-th epoch is used. We only consider the model after 10 epochs of training just to ensure that NBDT effect is there (previous epochs may have better scores). The model that we use attains an accuracy of 72.16% which is a 2.49% drop from the original class-labeling accuracy; compared to other results in table 2, there is nothing remarkable with the change in accuracy. The original NBDT paper also presents a result where NBDT causes a small accuracy drop in one of their models (and a small increase in another), so we believe this is not a crippling issue. However, this is significant when compared to the classification

Table 2: Class prediction accuracy. Apart from NBDT (ours), the other results are quoted from the original NBDT paper (Wan et al., 2021), including XOC (Alaniz & Akata, 2019), NofE (Ahmed et al., 2016).

| Method | NBDT (ours) | NBDT | NBDT | XOC | NofE |
|---|---|---|---|---|---|
| Backbone | ResNet50 | EfficientNet | ResNet18 | ResNet152 | AlexNet |
| Original Acc | 74.65% | 77.23% | 60.76% | 78.31% | 56.55% |
| Delta Acc | -2.49% | -0.63% | +0.50% | -17.5% | +4.7% |
| Explainable Acc | 72.16% | 76.60% | 61.26% | 60.77% | 61.29% |

accuracy obtained from WSOL training performed by (Choe et al., 2020). As shown in their section C2 appendix, WSOL training might cause classification accuracy to suffer e.g. prediction on CUB dataset may go as low as 21% in accuracy. Their ResNet50 classification performance is generally in the vicinity of 63%. Furthermore, we have shown that fine-tuning ResNet50 with NBDT is possible, which is much more efficient than the original NBDT paper that initialized weights from scratch and performed a very long 200-epoch training for ImageNet.

## 5 CONCLUSION

**Best values**. Naturally, it is easier to attain a correct localization with lower $\delta$. It might be debatable whether the average over different $\delta$ values is necessary or if the set of $\delta$ values should be lowered e.g. $10, 15, 30$ (at the risk of overestimating IoU performance). Regardless, we present our best findings: best score before NBDT is attained by Saliency method with the score +8.428 above CAM baseline at $\delta = 30$, while the best score post-NBDT is +10.688 above baseline, also at $\delta = 30$.

**Shortcomings, future directions and suggestions**. Repeating the experiments by sweeping through every possible variation of XAI methods for WSOL over different layers is not feasible. To prevent blind search of WSOL information hidden within many possible layers, future neural network architectures can be designed with special layers that are specifically taylored to perform WSOL, aimed at optimizing MaxBoxAcc metric. As we previously mentioned, a method with good MaxBoxAcc score can be very useful in terms of practicality (less cherry-picking of $\tau$). More efforts can be spent to maintain the accuracy of post-NBDT models so that interpretability, WSOL and predictive powers of the future models are all improved. The integration of XAI methods with the NBDT might be useful too: more research can be done to find a meaningful way to propagate signals in the XAI methods through the decision tree.

**Summary**. We have presented our investigations of (1) an existing WSOL metric that handles ill-posed localization problem (2) the different WSOL capability of various XAI methods at different layers. We have also repeated the entire experiment for the same model after NBDT training and compared their results. *Caveat*. Finally, we should mention that the technical and subtler details in the appendix are worth perusing. We have used the codes from two main papers (NBDT and Choe's) mostly as they are presented in their respective github repositories. There might be some concerning details that readers would like to pay attention to regardless of the fact that they have been published in top conferences.

### ACKNOWLEDGMENTS

Anonymous for now

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

## A  APPENDIX

### A.1  DATA AND SETUPS

Our codes are available in the supplementary materials and they are directly derived from the github repository of our two main references, namely Choe's paper https://github.com/clovaai/wsolevaluation and NBDT paper https://github.com/alvinwan/neural-backed-decision-trees. We strongly advise users to first set up everything specified in BOTH repository (including the arrangement of data). Make sure that both codes work before running our code.

In our code, the code base is Choe's github repository, and NBDT is included as a package (the folder is named 'nbdt'). The parts of codes that we write ourselves are clearly denoted with prefixes 'x' to distinguish them from the original codes. Instructions to run the codes are listed in README.md.

### A.2  TECHNICALITIES AND SUBTLER DETAILS

This paper is based on the first version of available codes. The next versions with minor fixes and clean-ups will be available after publication.

Table 3: Accuracy (%) by epoch. Epoch 12 (starred) is used for experiment.

| epoch | 0 | 1 | 2 | 3 | 4 | 5 | 6 | 7 | 8 | 9 |
|---|---|---|---|---|---|---|---|---|---|---|
| acc (%) | 72.32 | 72.02 | 72.24 | 72.50 | 72.57 | 71.91 | 72.0 | 72.29 | 72.22 | 72.09 |

| epoch | 10 | 11 | 12* | 13 |
|---|---|---|---|---|
| acc (%) | 72.06 | 71.96 | 72.16 | 71.56 |

### A.2.1 EVALUATION AND METRIC

**Codes**. As mentioned, our main references are Choe's paper (Choe et al., 2020) (CVPR 2020), the NBDT paper (Wan et al., 2021) (ICLR 2021). We use their evaluation codes as they are and followed their dataset settings.

**MaxBoxAcc is a stringent metric**. Why do we talk about how stringent is the metric? In Choe's paper, the CAM baseline result for models without WSOL finetuning is not presented anywhere, and we reiterate that our paper is intended partially to fill this gap. Because of the stark difference between the baseline we obtained (average 2.143 for our baseline) and their results which are fine-tuned for WSOL (around ~60), we are compelled to present readers a clearer picture regarding the MaxBoxAcc metric.

**MaxBoxAccV2.1**. To reinstate assumption (2), we set multi_contour_eval to False.

**Top 1 NBDT accuracy**. We follow the evaluation code of NBDT paper as is. The accuracy reported in the paper is computed on the validation dataset of ImageNet, so we do the same. We use their exact same code, The instructions for training and evaluation we followed are listed in their original README.md.

### A.2.2 ARCHITECTURE

In our code, we use ResNet50CAM, which is technically just ResNet50, but with additional function including CAM computation and create-split-at-layer() to perform layer splitting.

### A.2.3 NBDT TRAINING AND CHECKPOINTS

Pytorch DataParallel is for training across different GPUs. Despite the warning given on the main Pytorch page, we follow the original NBDT paper since their superior results are obtained using the said module. Some weaknesses of DataParallel include lost in-place updates, see `https://pytorch.org/docs/stable/generated/torch.nn.DataParallel.html`.

Main result is shown in table 3. Accuracy used is top-1.

**For authors' own reference**: We are limited by our GPU server queue. More precisely, every job can be submitted at a maximum of 48 hours, hence we need to split training across several jobs. To prevent repetitive, unproductive training, we save the checkpoint at each epoch and separately save the best result of that particular run rather than loading from the latest best checkpoint. Results with relabeling are shown in table 4. We notice there are minor mislabelings, e.g. if the latest epoch saved is epoch $n$, in the next run, each epoch is started at $n$ while it should have been $n + 1$. For now, we rectify and present them in the "relabeled epoch" column. Most importantly, the NBDT-trained checkpoint we used is checkpoint 9 before relabeling, or overall epoch 12 (after relabeling) with accuracy 72.16%.

Table 4: Please refer to table 3 for the main result. This is *only for authors' reference*: this table is included only for reference to a minor mislabelling issue (see explanation in appendix section A.2.3). Results as stored in pbsarxiv folder (left column). Starred epochs denote epochs where checkpoints are separately saved. epoch=12 is taken for experiment, i.e. the model after 13-th epochs of training.

| Run no. | epoch : acc(%) | relabeled epoch : acc(%) |
|---|---|---|
| 1 | 0*: 72.32 , 1:72.02 ,2: 72.24 | Same |
| 2 | 2*: 72.50 , 3*:72.57, 4:71.91 , 5:72.0 | 3*: 72.50 , 4*:72.57 , 5:71.91 , 6:72.0 |
| 3 | 5: 72.29*, 6: 72.22, 7: 72.09, 8: 72.06 | 7*: 72.29, 8: 72.22, 9: 72.09, 10: 72.06 |
| 4 | 8: 71.96, 9*: 72.16 (used) , 10: 71.56 | 11: 71.96, 12*: 72.16 (used) , 13: 71.56 |

