# OpenReview forum: "Evaluating Weakly Supervised Object Localization Methods Right? A Study on Heatmap-based XAI and Neural Backed Decision Tree"
_ICLR.cc/2023/Conference — Submitted to ICLR 2023_

### Official Review · Reviewer_8A57 · 2022-10-20

**Confidence:** 4
**Correctness:** 2
**Technical Novelty And Significance:** 1
**Empirical Novelty And Significance:** 1
**Recommendation:** 1

**Clarity, Quality, Novelty And Reproducibility:**

The manuscript is poorly presented. The arrangement of materials is chaotic. Language is poor and it's difficult to understand most of the materials.

This manuscript lacks novelty. The proposed method applies an existing training scheme to a well-known WSOD method.



**Strength And Weaknesses:**

In summary, this manuscript is of very bad quality.

Introduction:
- The research question is insufficiently motivated. What makes it worthwhile to improve the localization performance of XAI methods?
- What's the difference in definition between XAI methods and WSOD methods?
- "The NBDT original paper (Wan et al., 2021) trains larger models from scratch. However, we successfully fine-tuned pre-trained ResNet50 without causing the collapse of predictive performance." I don't see why this is a contribution.
- "Firstly, WSOL with only image-level labels can be an ill-posed problem." I don't see why.

Method & Experiment:
- Why choose MaxBoxAcc over mAP (detection) and Dice (segmentation)?
- Table 1, please report absolute performance.





**Summary Of The Paper:**

This manuscript aims to improve the localization performance of XAI models. The authors claim that incorporating NBDT into CAM can result in improved BoxAcc metric.

**Summary Of The Review:**

This manuscript resembles more a poorly-written school report than a research paper.

---

> ### Author Response · Authors · 2022-11-14
> **Our clarifications**
>
> Thank you for the comments and feedback, we will keep them in mind as part of our guides to improve the work. Here are some follow up clarifications.
>
> ## Introduction
> ---
> **“What makes it worthwhile to improve the localization performance of XAI methods?”**
>
> CAM has been one of the earlier successful XAI methods that are used for localization, helping us understand the prediction made by a black-box deep neural network. Many similar methods have emerged and improving localization as explanation may help improve the credibility of XAI methods in general.
>
> **“What's the difference in definition between XAI methods and WSOD methods”**
>
> The task is slightly different. A classifier model can function on its own, and XAI methods extract intermediate signals that the classifier uses for prediction. These intermediate signals are then formulated to create localization bounding boxes as the explanation to improve the transparency of a black-box models.
>
> WSOD, on the other hand, uses the black-box model as the core function to achieve object detection. XAI can be applied to a WSOD model to extract intermediate signals that can be interpreted as explanations. An example of how they differ: XAI may provide additional cues from the environment rather than just localizing the object.
>
> **“The NBDT original paper (Wan et al., 2021) trains larger models from scratch… I don't see why this is a contribution.”**
>
> We understand that to some people, the amount of resources spent is not an issue. However, saving 190 epochs of training has been an important part in our experiment.
>
> **"Firstly, WSOL with only image-level labels can be an ill-posed problem. I don't see why"**
>
> We apologize that this has not been very clear for the reader. As mentioned in our paper, this ill-posedness has been explained well in Choe et al, 2022, one of the two main references we use in our paper.
>
> ## Method & Experiment:
> ---
>
> **"Why choose MaxBoxAcc over mAP (detection) and Dice (segmentation)?”**
>
> While formulating our project, at first, we consider many different ways to quantitatively report localization results. Then we stumble upon Choe et al who explained that the usual WSOL measurements may have been ill-posed. Their arguments convince us to use their metrics. However, rather than using their metrics blindly, we decide to test it out first, therefore this paper is born.
>
> **"Table 1, please report absolute performance"**
>
> The absolute performance can be easily by adding the baseline 2.143 (on top of the table) to each number in the table.

---

### Official Review · Reviewer_53je · 2022-10-22

**Confidence:** 4
**Correctness:** 1
**Technical Novelty And Significance:** 2
**Empirical Novelty And Significance:** 2
**Recommendation:** 1

**Clarity, Quality, Novelty And Reproducibility:**

# Clarity and Quality
* I found the paper to be a rather confusing experience in its current form. The overall structure seems a bit disorganized. Perhaps more signposting would be helpful: "here's what we're going to do, here's why, here's what we found" etc.
* It would be better to use consistent notation in the presentation of CAM and MaxBoxAcc.
* There is a lot of "jargon dropping" without context or explanation - what should the reader take away from mentions of "mode collapse" or "embedded decision rules for soft inferences"? Please make sure to explain these terms and only include them if they contribute to the reader's understanding.
* [Page 2: "The problems with these simple metrics are well described in... First, WSOL with only image-level labels can be an ill-posed problem."] As presented, a reader might think that this is a motivation for the introduction of the MaxBoxAcc metric. Per Choe et al., this is not the case. My understanding is that ill-posedness of WSOL is brought up in Choe et al. to make it clear that bounding boxes are necessary in the validation set.

# Novelty
* The paper benchmarks a number of XAI methods on WSOL with and without NBDT, which is novel to the best of my knowledge.

# Reproducibility
* The section "CAM-inspired sum of weighted feature maps" is confusing, and does not seem to be described in sufficient detail for reproducibility.
* Are any hyperparameters tuned for this work? If so, how? If not, why is this not problematic for the experimental results in e.g. Table 1?



**Strength And Weaknesses:**

# Strengths
* WSOL is an interesting problem that may lead to label-efficient methods for predicting bounding boxes.
* Benchmarking new methods on WSOL is a worthwhile service to the community.
* I agree that MaxBoxAcc is a somewhat odd metric, and I appreciate critical thinking about its pros and cons. The paper's related discussion of thresholds is interesting.
* The paper considers a number of different XAI methods, and the idea to train using NBDT is interesting.
* It is interesting to see the qualitative results in Figure 1 showing that XAI methods can produce heatmaps that are unsuitable for WSOL.
* The visualizations of "cascading bounding boxes" in Figure 3 are interesting.
* The per-layer experiments in Figure 4 are interesting.

# Weaknesses

* I suspect there may be some fundamental misunderstandings about WSOL and in particular [Choe]. The paper makes reference to "WSOL training" and says that [choe] performs "a class agnostic training to optimize" for WSOL performance. I have read [choe] and this claim does not seem right to me. The typical approach in WSOL (and in [choe]) is to (i) train a CNN end-to-end on a multi-class classification task, with various bells and whistles (which is the main differentiator between WSOL methods) and (ii) generate boxes by using CAM, thresholding, and then picking a connected component to draw a box around. Nothing about the training process is "class agnostic" - the different WSOL methods in [choe] differ in their method for training the backbone, but all of them train multi-class classifiers. This seeming confusion is repeated on Page 6 ("However, their performance on MaxBoxAcc have not been reported.") where it is used to insinuate that [choe] did not report MaxBoxAcc results for methods "without WSOL training" because performance was low. I find this all very strange, and I am eager to hear a clarification.
* Related to the previous point, the paper says that in [choe] "models are fine-tuned specifically to improve WSOL performance" - this is not true to the best of my knowledge.
* The numbers reported in Table 1 are extremely low compared to [choe]. The paper mentions that "Vanilla CAM is equivalent to GradCAM applied to layer 4" - if so, then that result in Table 1 disagrees with the exact same result in [choe] by a large margin. Unless an excellent explanation is provided, implementation issues seem likely.
* The paper uses only one dataset, which is not standard in the WSOL literature. Most papers use CUB and ImageNet. [choe] introduces a third. [cole] introduces a fourth.
* The paper says that it "attempts to simultaneously maintain the model's predictive power as we perform WSOL" unlike the methods in [choe] which improve localization "at the expense of predictive performance". However, there is no comparison provided between the predictive performance of the proposed methods and the methods in [choe]. On Page 6, there is a claim that "final class label prediction accuracy results have not been reported" by [choe] after "WSOL training" because it "might have degraded significantly" - the numbers in question can be found at [choe-results], which is linked on the GitHub page for [choe]. In fact, the results from [choe] for ResNet-50 on ImageNet are similar to those in Table 2.
* Related to the previous point, I would caution against publicly suggesting that other authors deliberately omitted unflattering results without providing any evidence. One instance is described in the previous bullet point, and another instance is where the paper concludes: "Finally, we should mention that the technical and subtler details in the appendix are worth perusing. We have used the codes from two main papers (NBDT and Choe’s) mostly as they are presented in their respective github repositories. There might be some concerning details that readers would like to pay attention to regardless of the fact that they have been published in top conferences." This seems to ominously imply that there are significant flaws in those papers. When we turn to the appendix, there seems to be just a repeated allegations against [choe] based on what I perceive to be a deep misunderstanding of that paper, which I have addressed above. Identifying flaws in published work is absolutely crucial, but it should be done professionally, and with great care and attention to detail.

[choe]:

@inproceedings{choe2020evaluating,
  title={Evaluating weakly supervised object localization methods right},
  author={Choe, Junsuk and Oh, Seong Joon and Lee, Seungho and Chun, Sanghyuk and Akata, Zeynep and Shim, Hyunjung},
  booktitle={Proceedings of the IEEE/CVF Conference on Computer Vision and Pattern Recognition},
  pages={3133--3142},
  year={2020}
}

[choe-results]:

https://docs.google.com/spreadsheets/d/1O4gu69FOOooPoTTtAEmFdfjs2K0EtFneYWQFk8rNqzw/edit#gid=0

[cole]:

@article{cole2022label,
  title={On Label Granularity and Object Localization},
  author={Cole, Elijah and Wilber, Kimberly and Van Horn, Grant and Yang, Xuan and Fornoni, Marco and Perona, Pietro and Belongie, Serge and Howard, Andrew and Mac Aodha, Oisin},
  booktitle={European Conference on Computer Vision},
  year={2022}
}

**Summary Of The Paper:**

This paper considers the problem of weakly supervised object localization (WSOL), in which the goal is to accurately predict bounding boxes using only image-level labels at training time. The goal of this paper is to assess the performance of different explainable AI (XAI) methods (GradCAM, Saliency, DeepLIFT, GBP) for WSOL. The paper also asses the utility of neural backed decision tree (NBDT) training for the backbone and studies the utility of using different weighted sums of activation maps from different layers to produce localization heatmaps. All experiments are performed on ImageNet.

**Summary Of The Review:**

This paper addresses and interesting problem and undertakes a worthy benchmarking project. However, this work seems to be built on significant misunderstandings of prior work, and makes serious negative claims about prior work that are insufficiently supported. Of course it is possible that I am the one with a profound misunderstanding, in which case I hope to be corrected.

---

> ### Author Response · Authors · 2022-11-14
> **Thank you for your great remarks!**
>
> Thank you very much for the great and insightful feedback, and we will accordingly make the appropriate adjustments with your feedback in mind. For now, we will do our best to clarify any points you raised.
>
> ## Regarding weaknesses
> ---
> **“I suspect there may be some fundamental misunderstandings about WSOL and in particular [Choe].”**
>
> We agree that the point raised here definitely warrant more attention. Although you mention that ‘Nothing about the training process is "class agnostic"’, we believe that’s not quite accurate. Following Choe et al’s github codes, you will notice that there is specific WSOL training that is separate from the classification training, and this is the part that degrades the classification performance. The model is a multi-label classifier, but let us quote Choe et al 2020:
> 1. “We advocate the measurement of localization performance alone, as the goal of WSOL is to localize objects (§3.1) and not to classify images correctly”
> 2. “There are in general great fluctuations in the classification results (26.8% to 74.5% on CUB).”
> 3. the results of degraded classifcation performance can be found in section C2 of their appendix.
>
> Perhaps class agnostic is too strong a phrase and we apologize if it has created some confusion. In our revised version, we will:
> 1. quote Choe et al, 2020 to clarify the idea that classification performance may degrade in their pipeline (see our revised “Main objective of this paper”, point 2).
> 2. replace “class-agnostic” with something like “classification has degraded”.
>
> We apologize if the remarks in our paper seem like insinuations of some sort. Regardless, in this paper we just want to quantitatively clarify the following. If Choe et al pipeline is summarized as “WSOL training yields poor classification and good MaxBoxAcc”, we show that “without their WSOL training, models with good classification have poor MaxBoxAcc”.
>
> ---
> **"models are fine-tuned specifically to improve WSOL performance".**
>
> We also refer to the previous point. Yes, Choe et al have a specific training pipeline to improve localization w.r.t MaxBoxAcc. It is unnamed in the paper but do check their github. They simply refer to it as “WSOL training” and we refer to it as such in our paper.
>
> ---
> **The numbers reported in Table 1 are extremely low compared to [choe]**
>
> Great remark, that’s the point of this paper. To understand why, readers should come to terms with the previous two points before proceeding. One should first agree that there is a specific WSOL training that degrades classification performance (we used the term “class agnostic”, but we will rephrase it) and that the training is specifically aimed at optimizing the localization metric. Once this is agreed upon, we can understand why the numbers are extremely low: it is because we don’t use that WSOL training as we do not wish to degrade classification performance.
>
> We show that when the WSOL training proposed in Choe et al 2020 is not performed, the MaxBoxAcc scores are very low. We wanted NBDT training instead because we are working on explainable AI. NBDT is potentially great for classification performance and interpretability.
>
> ---
> **The paper uses only one dataset**
>
> Our study performs an experiment that sweeps across the layers of ResNet50 for different XAI methods, to be repeated for NBDT. In essence, we already perform Choe et al WSOL evaluation process multiple times, a resource intensive process overall. ImageNet should be well-known enough to represent the idea we want to deliver in the paper. We will repeat the experiment on different datasets in the future if we are able to secure more computing resources, unless the final feedback indicates that the idea is completely not worth pursuing.
>
> ---
> **‘The paper says that it "attempts to simultaneously…”’.**
>
> Thanks for pointing this out! We realize that we have been referring to the v1 of Choe’s paper for a while and the “final class label prediction accuracy” is indeed not found in the main text. However, the degraded prediction accuracy is found in C2 of the appendix of v2 as we mentioned above, and we have missed it. As a correction, we will include this in our revision.
>
> We do have our own classification results, found in Table 2 and compared to results in Wan et al. In our revised submission, we have added a comparison between these accuracies and the results in Choe et al. They have been added to the paragraph just before our conclusion section. The degraded classification accuracies obtained in Choe et al is around 63% for most WSOL methods on ResNet50. Our NBDT classification accuracy is 74.65%.
>
> === continued in the next comment ===

---

> ### Author Response · Authors · 2022-11-14
> **<<continued from the previous comment>>**
>
> ---
> **Related to the previous point, I would caution against publicly suggesting that other authors deliberately omitted unflattering results without providing any evidence**
>
> Thank you for this feedback, very greatly appreciated. Let us summarize the changes we will make in response to your suggestion.
> 1. The classification accuracy without WSOL training is indeed not reported. With this paper, we report the said accuracy, and that is a core part of our paper, so this cannot be removed. However, we rephrase it as such to prevent unnecessary unsavoury implication: “In this work, we fill the gap in the literature and show that the scores are very low.”
> 2. The degraded classification accuracy after WSOL training is found in C2. We apologize for missing out on it previously, and this will be properly added into the paper as the following “However, final class label prediction accuracy might have degraded as shown in section C2 of their appendix…”. The idea of class agnostic model remains unchanged, though we have rephrased it.
> 3. As for the “subtle” details, “There might be some concerning details…”, we apologize for the lack of clarity. However, we feel somewhat compelled to point them out.
> The part where “there seems to be just a repeated allegations against [choe]” is actually just a clarification (section A2.1). The subtle possible technical flaws can be found in the next sub-sections. For example, in our appendix A2.3, the main technical result in Wan et al is achieved using Data Parallelism in a previous version of Pytorch. If you notice, there are warning signs in that pytorch page, e.g. regarding the erroneous batch norm when multi-GPU training is used. Interestingly, good results are still obtained in the paper, and when we repeat the experiment as it is, reasonably similar result can be obtained.
>
> We do need your advice to decide whether we should remove them from our paper before we submit our final revision.
>
> ---
> ## Clarity, Quality, Novelty And Reproducibility
> ---
>
> **“Perhaps more signposting would be helpful”**. Noted. Thanks, we will try our best to present the paper with a better flow. For example, all the introductions/topic sentences in the subsections of our “Methods” section have been revised.
>
> **“It would be better to use consistent notation in the presentation of CAM and MaxBoxAcc”**. Yes, we understand that GradCAM and CAM might raise some confusion. CAM is GradCAM at layer 4, but we do perform experiments on other layers with GradCAM, so we apologize that the whole notations cannot be more consistent (unless perhaps we change all CAM to GradCAM layer 4, which doesn’t seem sensible). As for MaxBoxAcc, yes, there is BoxAcc plot that is different from MaxBoxAcc. This is also present in Choe et al, 2020.
>
> **"There is a lot of "jargon dropping" without context or explanation"**. We’re really sorry for the jargon dropping. For “embedded decision rules for soft inferences”, we assume that readers have familiarzied themselves with NBDT paper (Wan et al). As for mode collapse, we believe it’s a standard term to describe training that fails (often one that results in a very skewed classification/prediction).
>
> **“As presented, a reader might think that this is a motivation for the introduction of the MaxBoxAcc metric”**. I believe the motivation is true. MaxBoxAcc is defined bottom up, part by part in a way that precisely tackles some of the problems they described e.g. “To fix the issue, we propose new evaluation metrics that are…”, quoted from Choe et al page 4. A few other problems they tried to address include WSOL researchers reporting validation results, and thus, as you pointed out, their system is designed “make it clear that bounding boxes are necessary in the validation set.”
>
> === continued in the next comment ===

---

> ### Author Response · Authors · 2022-11-14
> **<< continued from the previous comment >>**
>
> ---
> ## Reproducibility
>
> ---
>
> **‘The section "CAM-inspired sum of weighted feature maps" is confusing, and does not seem to be described in sufficient detail for reproducibility’.**
>
> For reproducibility, everything is available in the supplementary materials, including all formulas. The sum of weighted feature maps are really just a modification of CAM. CAM can be succinctly presented in a simple formla in terms of $w_{ik}$ (see page 2). However, empirically, the weights do not always work for other XAI methods, hence we perform several trial and errors to find the most qualitatively sensible $w_{ik}$, as described in the main text.
>
> ---
>
> **Are any hyperparameters tuned for this work? If so, how? If not, why is this not problematic for the experimental results in e.g. Table 1?**
>
> In the WSOL evaluation, there are no hyperparameters because we intend to replicate the WSOL metric results, expect without the WSOL training (because we do not want the classification performance to degrade). There is a minor hyper-parameter tuning in our NBDT training. This is because we do not wish to train ResNet from scratch like Wan et al did. Luckily, with only a few trials, we obtained the parameters required to fine-tune pretrained ResNet50 without causing collapse (available in the main text, also available in supp. materials).

---

> ### Comment · Reviewer_53je · 2022-12-12
> **Still recommending rejection**
>
> I'd like to thank the authors for their engagement! Ultimately I still recommend rejection for this paper.
>
> My major remaining issues are as follows:
> * The extended discussion above (https://openreview.net/forum?id=X55dLasnEcC&noteId=8_5EzHZYPbr) helped to clarify a few things, but we ultimately agreed that there are some large and unexplained discrepancies between the results in this work and those from Choe et al. that need to be thoroughly understood. Without more investigation, we can't be sure that implementation issues aren't the source of the discrepancy.
> * The experiments are based on only one dataset (ImageNet). Most WSOL papers also study CUB at minimum, but others (OpenImages30k, iNatLoc500) are available as well. The authors raise a valid point about limiting computational cost, but all of these datasets are much smaller than ImageNet and so the computational burden would be relatively light.
> * There are a number of points where I consider the paper to be misleading in its current form (e.g. the use of the terms "class agnostic" and "WSOL training" -- see the discussion linked above). I would want to see a rewritten version of the paper before changing my rating.
>
> While I think the goal of the paper is worthwhile (e.g. connecting XAI methods to WSOL), I think the paper needs more work before it's ready.

---

> > ### Author Response · Authors · 2022-12-12
> > **Can't wait to get the official rejection too**
> >
> > It has been a great discussion, no doubt! We really appreciate the great lengths reviewer 53je is willing to go through with us.
> >
> > Just a few minor comments from us,
> > 1. "Without more investigation, we can't be sure that implementation issues aren't the source of the discrepancy." It's absolutely true. The only meagre thing we can say in our own defense is that we've tried to run the code with as little modification as possible from the original code.
> > 2. "The experiments are based on only one dataset (ImageNet)..." We will keep this in mind for our future works.
> > 3. In the revised version (we have submitted it), all reference to "class agnosticity" has been replaced with something along the line of "classification performance has degraded", including supporting quotes like "There are in general great fluctuations in the classification results (26.8% to 74.5% on CUB)" etc. As for the use of the term "WSOL training", we refuse to change it because it is the exact term that Choe et al uses in their github. We do not wish to confuse readers with multiple terms.

---

### Official Review · Reviewer_zRUD · 2022-10-24

**Confidence:** 5
**Correctness:** 3
**Technical Novelty And Significance:** 1
**Empirical Novelty And Significance:** 1
**Recommendation:** 1

**Clarity, Quality, Novelty And Reproducibility:**

+ The paper is clearly written and organized.
+ While WSOL methods are important for machine learning models, in the context of interpretability and explainability, this paper fails to provide a novelty or contribution to the field.

**Details Of Ethics Concerns:**

None.

**Strength And Weaknesses:**

Strength:
+ The paper concerns WSOL tasks and explainability, which are very important and relevant.
+ The empirical results show some benefits of the method, and some ablation studies are provided.

Weakness:
+ There is a lack of contribution and novelty.


**Summary Of The Paper:**

In this paper, the authors discussed the work of (Choe et al., 2020) for the evaluation of WSOL methods. and the work of (Wan et al., 2021)  for the neural-backed decision tree (NBDT) method. The authors describe both works, and also evaluate some heatmap-based explainable methods such as GradCAM, saliency, Deeplift, and Guided Backpropagation, and compare them to the CAM method. They also compare with and without NBDT training. Evaluation is done on the Imagenet dataset.

**Summary Of The Review:**

+ This paper reads more like a tutorial rather than a research paper. It lacks any innovative contribution. In addition, it is not clear why authors refer to WSOL methods (Choe et al 2020) as class-agnostic - they are class-aware methods.

---

> ### Author Response · Authors · 2022-11-14
> **A Fresh Perspective to MaxBoxAcc**
>
> Thank you for your time! Indeed, we are discussing the WSOL evaluation method proposed by Choe et al, 2020 and NBDT (Wan et al, 2021).
>
> ---
>
> **Regarding the weaknesses of this paper**
>
> Regarding the “lack of contribution and novelty,” we hope that you could provide a little more guidance on how to improve the paper. We believe we do have some new perspectives into MaxBoxAcc, an interesting existing localization metric. Here are some points that may be of interest:
>
> 1. The performance (BoxAcc) vs threshold in figure 4 shows patterns we believe have never been seen before. For example, the spike in saliency’s threshold is perhaps an unusual feature.
> 2. Also, we point out that baseline CAM has not yielded good MaxBoxAcc scores without the special WSOL training by Choe et al, 2020. On the other hand, with WSOL training, classification performance degrades. This presents a great opportunity for explainable AI: if we can improve classification performance while scoring high on MaxBoxAcc metric, we will obtain a more robust WSOL system, especially so because MaxBoxAcc has been defined with a good intention to solve the ill-posedness problem.
> 3. Finally, we provide explanations on why MaxBoxAcc score can be so low even in the baseline case without WSOL training. In the main content of our paper, we particularly explain how the small window of correct threshold is needed to achieve good MaxBoxAcc scores. Perhaps future researchers could start their designs with this consideration in mind.
>
> ---
>
> **Regarding the summary**
>
> “In addition, it is not clear why authors refer to WSOL methods (Choe et al 2020) as class-agnostic - they are class-aware methods”. Choe et al 2020 paper itself suggests that their model is class agnostic after training. We quote their paper:
> 1. “We advocate the measurement of localization performance alone, as the goal of WSOL is to localize objects (§3.1) and not to classify images correctly”
> 2. “There are in general great fluctuations in the classification results (26.8% to 74.5% on CUB).”
> 3. the results of degraded classifcation performance can be found in section C2 of their appendix.
>
> Perhaps class agnostic is too strong a phrase and we apologize if it has created some confusion. In our revised version, we will
> 1. quote Choe et al, 2020 to clarify the idea that classification performance may degrade in their pipeline (see our revised “Main objective of this paper”, point 2).
> 2. replace “class-agnostic” with something like “classification has degraded”.

---

### Official Review · Reviewer_GA9G · 2022-10-28

**Confidence:** 1
**Clarity, Quality, Novelty And Reproducibility:** The proposed method in this paper is …
**Correctness:** 2
**Technical Novelty And Significance:** 2
**Empirical Novelty And Significance:** 2
**Recommendation:** 3

**Strength And Weaknesses:**

This paper finds that XAI methods perform WSOL with very sub-standard MaxBoxAcc scores.

The experiment is then repeated for the same model trained with Neural Backed Decision Tree (NBDT) and finds that vanilla CAM yields significantly better WSOL performance after NBDT training.

**Summary Of The Paper:**

This paper measures the WSOL capability of existing heatmap-based XAI method applied on ResNet50 and improves them.

**Summary Of The Review:**

The overall contributions of this paper are not significant.

---

> ### Author Response · Authors · 2022-11-14
> **Thank you**
>
> Thank you for the review! The findings in our paper have been described summarily.
>
> “The proposed method in this paper is incremental”. Our overarching goal is to provide a result for researchers to consider MaxBoxAcc metric in a different light. The metric has a very good potential (it solves ill-posedness problem), but our paper shows that more can be done in the WSOL field to achieve high scores without sacrificing classification performance.

---

### Comment · Reviewer_53je · 2022-11-16
**The WSOL metric in Choe et al. in class-agnostic, the WSOL methods are not.**

I'm making a brief top-level post because I think this is a key point to clarify.

My understanding is as follows:
1. The MaxBoxAccV2 **metric** is class-agnostic.
2. The WSOL **methods** considered in Choe et al. are not class agnostic. In particular, they are trained for multi-class classification.

Authors, please let me know if you disagree with either of these statements - this is a fundamental point and we need to arrive at consensus.

---

> ### Author Response · Authors · 2022-11-16
> **Partially agree**
>
> Thanks for this clarification thread!
>
> As far as we understand, MaxBoxAcc is a WSOL metric, a more sophisticated variant of Dice or top-1. At the core it is measuring hit/miss of bounding boxes regardless of the class, so yes it's class agnostic.
>
> Each underlying model is not class agnostic because ResNet etc have output that can perform classification.
>
> However, the nameless WSOL method (referred to simply as 'WSOL training' in their github) is used to train the model further so that MaxBoxAcc is optimized i.e. so that the model can now perform localization well. As a result, while the model is technically still capable of producing classification labels, the performance has degraded. Some models do degrade to the point of 20% classification accuracy, that's why in our first version we call this training process class agnostic.

---

> > ### Comment · Reviewer_53je · 2022-11-16
> > **Getting to the bottom of the "nameless WSOL method"**
> >
> > Thanks for the quick reply, I think we're getting to the bottom of this.
> >
> > > However, the nameless WSOL method (referred to simply as 'WSOL training' in their github) is used to train the model further so that MaxBoxAcc is optimized i.e. so that the model can now perform localization well.
> >
> > My understanding is that Choe et al. consider several WSOL methods (CAM, HaS, ACoL, SPG, ADL, CutMix) which all differ in the way they  train a multi-class image classifier (whose feature maps will ultimately be repurposed for object localization). Is your claim that there is a "nameless WSOL method" used in Choe et al. that is distinct from these techniques? Can you provide a pointer to the place in the code of Choe et al. where this method is implemented?

---

> > > ### Author Response · Authors · 2022-11-16
> > > **I see, there's multiple layers of things in Choe's paper**
> > >
> > > Now I see where is the point of confusion. There is a sequence of different things involved in Choe's paper and probably we've focused on different parts for a while.
> > >
> > > CAM, HaS, ACoL, SPG, ADL, CutMix are the final mechanism to produce the localization bounding boxes in Choe's paper. But before they can be used to score reasonably well on the MaxBoxAcc, the nameless WSOL training is a required intermediate process. They are all essential part of the process in the paper.
> > >
> > > [edit] As for the code it is described here in section 6 of the main github page https://github.com/clovaai/wsolevaluation#6-wsol-training-and-evaluation.
> > >
> > > Go to the main.py, see line 371, train_performance =  trainer.train(split='train'). The class is defined in line 62, the function in line 191. Inside this function, there's a _wsol_training too, and you can trace further.

---

> > > > ### Comment · Reviewer_53je · 2022-11-16
> > > > **This might be the root of the issue**
> > > >
> > > > I believe your understanding is incorrect. CAM, HaS, ACoL, SPG, ADL, and CutMix are not a "final mechanism to produce the localization bounding boxes" - instead, they are techniques for training image classifiers. The notion is that by training your image classifier in certain ways, you'll get feature maps that are more useful for downstream localization. But the end product is just a function that takes an image and produces feature maps.
> > > >
> > > > As we can see here (https://github.com/clovaai/wsolevaluation/blob/master/main.py#L164), the "WSOL training" is simply choosing which of the methods (CAM, HaS, ACoL, SPG, ADL, CutMix) to use.
> > > >
> > > > As we can see here (https://github.com/clovaai/wsolevaluation/blob/master/main.py#L271), the final method used to produce the localization bounding boxes is always class activation mapping followed by thresholding and connected component finding. There is no special "final mechanism to product localization bounding boxes" other than this.
> > > >
> > > > The term "CAM" is used in two senses in Choe et al. First, it is the method for getting from feature maps to bounding boxes, as described above. Second, it is the name used to describe the baseline WSOL method that simply does no special tricks during classifier training to improve the feature maps.
> > > >
> > > > Please let me know where you disagree.

---

> > > > > ### Author Response · Authors · 2022-11-16
> > > > > **You're right but "final mechanism" is not wrong**
> > > > >
> > > > > No, i don't disagree. The WSOL training is indeed the wrapper that eventually runs the whole pipeline of this localization optimization training, and the little _wsol_training() function is the specific part that chooses the specific schemes out of those 6. In Choe's paper, the training is necessary** to attain the good MaxBoxAcc score.
> > > > >
> > > > > Each of the 6 methods IS a heatmap generation method, which is later processed a little further with python cv2 functions to obtain the bounding boxes. But yes you can say that they are intermediate modifications before the original CAM method. In any case, since the signals they propagate forward is a necessary part to extract the attention map, as a whole they are THE bounding box generation methods. So, yes, they are also the final mechanism to get the bounding box, if you are using them for computing MaxBoxAcc (and not for WSOL training). I think arguing around this is just arguing around different perspectives.
> > > > >
> > > > > **Just repeating a previous point here: the whole point of our paper is we don't perform any of this WSOL training i.e. none of those 6 methods (5 actually). We want to see if CAM, on its own, can perform good localization with MaxBoxAcc metric. Why? Again, because we're doing XAI, and we want to preserve the classification power and extract localization as explanation, which is unlikely to be possible with their WSOL training (as it degrades classification performance).

---

> > > > > > ### Comment · Reviewer_53je · 2022-11-16
> > > > > > **Isn't CAM from Choe et al. what you're describing?**
> > > > > >
> > > > > > Thanks again for the quick response, this is helpful!
> > > > > >
> > > > > > I certainly don't want to argue a semantic point, but I don't actually think that's what we're looking at here.
> > > > > >
> > > > > > > Just repeating a previous point here: the whole point of our paper is we don't perform any of this WSOL training i.e. none of those 6 methods (5 actually). We want to see if CAM, on its own, can perform good localization with MaxBoxAcc metric. Why? Again, because we're doing XAI, and we want to preserve the classification power and extract localization as explanation, which is possible with their WSOL training (as it degrades classification performance).
> > > > > >
> > > > > > I believe the "CAM" method in Choe et al. (i.e. the WSOL method, not the component of the box generation pipeline - unfortunate about the confusing terminology!) does exactly what you're saying. There is no specialized WSOL training at all - just plain old multi-class classification training. Do we agree on that?

---

> > > > > > > ### Author Response · Authors · 2022-11-16
> > > > > > > **Let us express our appreciation first**
> > > > > > >
> > > > > > > Actually we're the one very thankful for your effort to go through all these haha. This is the most detailed effort I (the first author) got so far in peer review (our rating is so low right now it's probably not worth your effort to go through this hassle). No offense ever meant in any of our comments btw; if any was felt, it's not intended.
> > > > > > >
> > > > > > > There is a specialized training. In Choe et al, with the plain old CAM, the _wsol_training still forward the following objects that carry gradients.
> > > > > > > ```
> > > > > > > output_dict = self.model(images, target)
> > > > > > > logits = output_dict['logits']
> > > > > > > ```
> > > > > > > This is part of a whole training, that's why there is a localization optimization in their paper. We don't use this at all and try to measure the MaxBoxAcc of original CAM (which is only the weighted sum of feature maps part that generate heatmaps). Then, we use NBDT training in our second part of the paper and again measure the accuracy.

---

> > > > > > > > ### Comment · Reviewer_53je · 2022-11-16
> > > > > > > > **Still not clear on how that's specialized training?**
> > > > > > > >
> > > > > > > > Of course, you're very welcome! No offense taken.
> > > > > > > >
> > > > > > > > I'm still not convinced that there is any specialized training.
> > > > > > > >
> > > > > > > > My understanding is that the CAM method in Choe et al. consists of two stages:
> > > > > > > > 1. Train time: Train a multi-class classifier in a completely vanilla way (image -> logits -> categorical cross-entropy against class labels, backprop, repeat).
> > > > > > > > 2. Test time: Use CAM + thresholding + connected component finding to generate bounding boxes.
> > > > > > > >
> > > > > > > > I am very eager to know what steps you think Choe et al. are taking during their CAM method, if not what I've outlined above. Please be as precise as you can, ideally with a link to lines of code. Where can I find this "localization optimization in their paper"? The code you copied above is just showing that Choe et al. use the logits for something -- in particular, they are fed into the categorical cross-entropy loss during (what I understand to be completely vanilla) multi-class classifier training. This is consistent with my claim that the CAM approach does not involve any "localization optimization" during training.

---

> > > > > > > > > ### Author Response · Authors · 2022-11-16
> > > > > > > > > **"Train time" might be the key**
> > > > > > > > >
> > > > > > > > > The "train time" is not to train multi-class classifier. If you notice their github, the class label/classifier training is long past: they use PRETRAINED_MODEL=True i.e. the label classification training is already done and out of question. See this in section 6 of https://github.com/clovaai/wsolevaluation after "Below is an example command line for the train+eval script."
> > > > > > > > >
> > > > > > > > > Whatever training happening there starting from the github main.py line 191 is WSOL training for better localization.
> > > > > > > > >
> > > > > > > > > By contrast, the starting point in our paper is from the pretrained model.
> > > > > > > > >
> > > > > > > > > [edit] I think i know what's the most convincing way to say this. To answer 'Where can I find this "localization optimization in their paper"?', it's main.py line 264 evaluate(). In the case of CAM, they continued training as usual with logits to minimize classification loss. The difference is which checkpoint in the model they eventually pick. They eventually picked the checkpoint that does localization best, determined through their evaluation process. So they optimize the localization by selecting the best checkpoint.

---

> > > > > > > > > > ### Comment · Reviewer_53je · 2022-11-16
> > > > > > > > > > **Think we're reaching common ground!**
> > > > > > > > > >
> > > > > > > > > > Yes, they start from ImageNet pretraining and then train for 10 more epochs using the standard classification loss. But according to (https://github.com/clovaai/wsolevaluation):
> > > > > > > > > >
> > > > > > > > > > > Note that we evaluate the last checkpoint of each training session.
> > > > > > > > > >
> > > > > > > > > > And according to (https://arxiv.org/pdf/2001.07437.pdf):
> > > > > > > > > >
> > > > > > > > > > > After the acceptance by CVPR 2020, we believe that it is inappropriate to use the best checkpoint for WSOL evaluation. This is
> > > > > > > > > > because the best localization performances are achieved before convergence in many cases...
> > > > > > > > > >
> > > > > > > > > > > We re-evaluate six recently proposed WSOL methods with MaxBoxAccV2, and the results are shown in Table 8. All training configurations but evaluation metrics are the same as those of the main paper. We evaluate the last checkpoint of each training session. We obtain the same conclusions as with the original metric, MaxBoxAcc: (1) there has been no significant progress in WSOL performances beyond vanilla CAM [60] and (2) with the same amount of fully-supervised samples, FSL baselines provide better performances than the existing WSOL methods.
> > > > > > > > > >
> > > > > > > > > > How is your claim that "CAM (baseline) MaxBoxAccV2.1 = 2.143" consistent with their newer results, given that they are no longer using the checkpoint that does localization best but still claim high performance? I know you might have originally considered their old results, but their "new" results were published back in 2020 so we need to consider them as well.

---

> > > > > > > > > > > ### Author Response · Authors · 2022-11-17
> > > > > > > > > > > **MaxBoxAcc V2 assumption 2**
> > > > > > > > > > >
> > > > > > > > > > > It's somewhat surprising to us how they achieve such high scores for the case of CAM. To summarize what we know so far: we don't perform any WSOL training. They do, but in their revised version, only the last checkpoint is evaluated. In the case of CAM, even with standard classification loss training, their MaxBoxAccV2 performance is still good.
> > > > > > > > > > >
> > > > > > > > > > > There is an important difference that can result in the low score of our paper that we can think of. Our 2.143 baseline is the result we obtained on MaxBoxAcc V2.1, where assumption (2) of their MaxBoxAcc is reintroduced to the metric. The assumption of interest is in Choe's paragraph just before section 4.2:
> > > > > > > > > > > ```
> > > > > > > > > > > MaxBoxAcc takes the *largest* connected component for estimating the box, assuming that the object of interest is usually large. We remove this assumption by considering the best match between the set of all estimated boxes and the set of all ground truth boxes
> > > > > > > > > > > ```
> > > > > > > > > > > Since their new metric considers a match with the ground-truth, it's probably natural that they score better. We reinstated this assumption in V2.1 with a reasonably sensible justification mentioned in our paper
> > > > > > > > > > > ```
> > > > > > > > > > > a good XAI method is expected to yield accurate heatmaps/attributions in which large noisy patches should be suppressed or not exist altogether i.e. there is no large artifact. The most salient box for an accurate heatmap is thus already the largest component area
> > > > > > > > > > > ```
> > > > > > > > > > > In a sense, we're trying to measure the localization performance of good quality heatmaps rather than ones that are matched to ground-truth.

---

> > > > > > > > > > > > ### Comment · Reviewer_53je · 2022-11-19
> > > > > > > > > > > > **Important to figure out the reason for the discrepancy**
> > > > > > > > > > > >
> > > > > > > > > > > > Got it. If the claim is that MaxBoxAccV2 and MaxBoxAccV2.1 lead to scores that differ so vastly, I think it's important to get to the bottom of that and clearly explain the root of that difference with copious evidence. This serves to rule out the possibility of implementation issues, which can be fiendishly hard to locate in large ML projects.

---

> > > > > > > > > > > > > ### Author Response · Authors · 2022-11-19
> > > > > > > > > > > > > **Agreed**
> > > > > > > > > > > > >
> > > > > > > > > > > > > Alright, agreed. For now, with this thread at least readers will be made more aware of the existence of differences in implementation. Thanks again for your feedback!

---

> > > > > > > > > > > > > > ### Comment · Reviewer_53je · 2022-11-19
> > > > > > > > > > > > > > **Thanks!**
> > > > > > > > > > > > > >
> > > > > > > > > > > > > > Of course, thank you for the constructive engagement!

---

### Decision · Program_Chairs · 2023-01-20

**Decision:**

Reject

**Justification For Why Not Higher Score:**

There are significant technical flaws in the work in its current form, as described in the meta-review, and the majority of reviewers also recommend strong reject.

**Justification For Why Not Lower Score:**

N/A

**Metareview: Summary, Strengths And Weaknesses:**

All reviewers agree that while this work tackles an important problem, there are some significant concerns and flaws in the current version of the work, including the characterization of WSOL in prior related methods. The contribution of this paper is also considered not significant enough to be publishable in ICLR at this time. This paper is therefore recommended to be rejected. The reviewers and authors have engaged in an active discussion, and we hope the authors will find this useful and can incorporate the feedback into revising and resubmitting the work elsewhere.